# Designing an Oxygen Scavenger Multilayer System Including Volatile Organic Compound (VOC) Adsorbents for Potential Use in Food Packaging

**DOI:** 10.3390/polym15193899

**Published:** 2023-09-27

**Authors:** Carol López-de-Dicastillo, Gracia López-Carballo, Pedro Vázquez, Florian Schwager, Alejandro Aragón-Gutiérrez, José M. Alonso, Pilar Hernández-Muñoz, Rafael Gavara

**Affiliations:** 1Packaging Group, Institute of Agrochemistry and Food Technology IATA-CSIC, Av. Agustín Escardino 7, 46980 Paterna, Spain; clopezdedicastillo@iata.csic.es (C.L.-d.-D.);; 2Evonik Operations GmbH, Germany; 3Grupo de Tecnología de Envases y Embalajes, Instituto Tecnológico del Embalaje, Transporte y Logística, ITENE, Unidad Asociada al CSIC, calle de Albert Einstein 1, 46980 Paterna, Spain

**Keywords:** oxygen scavenger, active packaging, food packaging, multilayer film, volatile organic compound, VOC

## Abstract

Oxygen scavengers are valuable active packaging systems because several types of food deterioration processes are initiated by oxygen. Although the incorporation of oxygen scavenger agents into the polymeric matrices has been the trend in recent years, the release of volatile organic compounds (VOC) as a result of the reaction between oxygen and oxygen scavenger substances is an issue to take into account. This is the case of an oxygen scavenger based on a trans-polyoctenamer rubber (TOR). In this work, the design of an oxygen scavenger multilayer system was carried out considering the selection of appropriate adsorbents of VOCs to the proposed layer structure. Firstly, the retention of some representative organic compounds by several adsorbent substances, such as zeolites, silicas, cyclodextrins and polymers, was studied in order to select those with the best performances. A hydrophilic silica and an odor-adsorbing agent based on zinc ricinoleate were the selected adsorbing agents. The principal VOCs released from TOR-containing films were carefully identified, and their retention first by the pure adsorbents, and then by polyethylene incorporated with the selected compounds was quantified. Detected concentrations decreased by 10- to 100-fold, depending on the VOC.

## 1. Introduction

Active packaging is an emerging food technology based on a deliberate interaction of the packaging with the product and/or its headspace to improve food quality and safety. Active packaging refers to those materials intended to interact with the internal gas environment and/or directly with the product with a beneficial outcome. Some of these newly employed technologies modify the gas environment by removing gases from or adding gases to the package headspace [1,2]. Oxygen scavengers are one of the most interesting active packaging systems since the presence of oxygen is known to trigger many food deterioration reactions, such as lipid oxidation, color changes, nutrient losses and microbial growth [1,2,3]. In comparison with other packaging technologies, such as Modified Atmosphere Packaging (MAP) and vacuum packaging, oxygen scavengers may greatly reduce the oxygen level in the headspace to less than 0.1 vol%, leading to an extended shelf life. In most commercial applications, oxygen-adsorbing substances have been included in sachets that are inserted into the package or as adhesive labels bonded to the inner wall of the package [4,5,6,7]. Nevertheless, this system presents some disadvantages, such as a possible accidental ingestion or misuse by the consumer, consumer rejection, and the need of an additional packaging operation to insert the sachet in each package. Therefore, although this technology is well established, these inconveniences have been the driving force of more recent developments, such as closure liners containing the oxygen scavenger, dissolution or dispersion of the active substances in the plastic material [8]. In this regard, designing functional polymeric materials that include the active agents in their structure and ensuring that these active substances do not cause relevant effects on packaging functional properties are both key factors. Metal and metal derivatives, such as finely divided palladium, cobalt or iron, and organic and fatty acids have been broadly used as active substances in the development of oxygen scavengers [3,4,6,9]. Regarding the polymeric material for use in an oxygen scavenging composition, this should exhibit good processing characteristics, be able to be formed directly into useful packaging materials, or have high compatibility with those polymers commonly used in food packaging designs.

Several research works and patent applications have disclosed that ethylenic-unsaturated hydrocarbons, such as squalene, fatty acids, or polybutadiene, present sufficient commercial oxygen scavenging capacity to extend the shelf life of oxygen-sensitive products [9,10,11]. These unsaturated hydrocarbons, after being functionally terminated with a chemical group to make them compatible with the packaging materials, can be added during conventional melt–mixing processes to thermoplastics, such as polyesters, low-density polyethylene (LDPE) and polypropylene (PP), and the films can be obtained using most conventional plastic-processing techniques, such as co-injection or co-extrusion. 1,2-polybutadiene is especially preferred thanks to its transparency, mechanical properties and processing characteristics similar to those of polyethylene [9,10,12]. The main problem of this technology is that during the reaction between these polyunsaturated macromolecules and oxygen, byproducts such as organic acids, aldehydes, or ketones can be generated, which are volatile organic compounds (VOCs) that can affect the sensory quality of the food or raise food regulatory issues [4,9]. This problem can be minimized through two strategies: (i) the use of functional barriers located between the food product and the scavenger layer that impede the migration of undesirable oxidation products, but allow oxygen transfer [13]; and (ii) the use of VOC adsorbent materials, either polymers with inherent organic compound-scavenging properties or the incorporation of adsorbers within the polymer structure (i.e., silica gel, zeolites, etc.).

In the present work, the oxygen scavenger component is based on trans-polyoctenamer rubber (TOR) in combination with a catalyst with several characteristics that made this product attractive for food packaging applications. Because the release of VOCs, such as organic acids, aldehydes and ketones, can become a drawback, the design of an oxygen scavenger multilayer system was carried out from the selection of appropriate adsorbents of VOCs to the proposal of the multilayer structure.

Therefore, the aim of this study was the reduction of VOCs released by TOR-containing films into the package headspace through the incorporation of a functional barrier with VOC adsorbents in the packaging system. Different adsorbing substances, such as zeolites, cyclodextrines and polymers, were tested in order to select and later develop the multilayer packaging system. The necessary milestones for the development of an efficient commercial prototype were detailed, from the selection of efficient adsorbents to the processing of the resulting multilayer system.

## 2. Materials and Methods

### 2.1. Materials

Films from TOR masterbatch (TM) were prepared and supplied by Evonik as experimental test specimens. These films, 50 µm low-density polyethylene (LDPE) film containing 10%wt. of TOR and tri-layer 10/10/10 µm (passive/active/passive) LDPE film containing 10%wt. of TOR in the active layer were supplied in high barrier Aluminum/LDPE bags under vacuum until use.

Adsorbents (supplier): Sylysia (Syl) SY350 and Dumacil (Duma) 100 FG K (Fuji Sylysia, Bussi, Italy), Tego Sorb PY 88 TQ (Tego) (zinc ricinoleate) (Evonik, Essen, Germany), β-cyclodextrin (βCD) and hydroxypropyl-β-CD (HPβCD), commercially named as Cavasol W7 and Cavamax W7, respectively (Wacker Chemie AG, Munich, Germany), Selvol Ultiloc 5003 (polyvinyl alcohol/polyvinyl amine) (Sekisui, La Canonja, Spain) and Tenax TA 60–80 mesh (modified polyphenylene oxide) (Merck Life Science S.L.U., Madrid, Spain).

Formic acid, acetic acid, hexanoic acid, acetaldehyde, propanal, valeraldehyde, hexanal, acetone, amyl formate and cyclooctane were purchased from Merck Life Science S.L.U. (Madrid, Spain).

AGILITY EC 7000 Performance LDPE, 0.919 g/cm^3^, melt index 3.9 g/10 min, and maleic anhydride grafted polyethylene used as tie layer Amplify TY-1057-H were purchased from Dow Chemical (Barcelona, Spain). Ethylene vinyl alcohol copolymer (EVOH) Eval F171B, metallocene linear low-density polyetlylene (mPE) Supeer 8115 and polypropylene (PP) Moplen RP 310 were supplied by Kuraray-Eval Europe (Zwijndrecht, Belgium), Sabic (Cartagena, Spain) and Lyondell Basell (Vilaseca, Spain), respectively.

### 2.2. Selection of Adsorbents

#### 2.2.1. Analysis of Retention Capacity of Adsorbents

Formic acid, acetic acid, hexanoic acid, acetaldehyde, propanal, valeraldehyde, hexanal, acetone, amyl formate and cyclooctane were selected as representative volatile organic compounds (VOCs) to carry out the evaluation of the adsorption capacity of adsorbents.

Fifty mg samples of each specific adsorbent were placed in a 22 mL vial, and the vial was hermetically sealed with an aluminum cap and butyl rubber/polytetrafluoroethylene (PTFE) septum. In parallel and for each organic compound, a known volume of the VOC was injected in a 120 mL glass vial closed with a butyl rubber/PTFE septum previously heated at 150 °C. The substance was allowed to evaporate for 30 min and then a calculated amount of the headspace was withdrawn and injected in the 22 mL vial with the adsorbent, to put quantities in the range 20–100 µg/vial in gas phase. These samples were allowed to equilibrate during 24 h at 23 °C. Then, 1 mL of the headspace was withdrawn and injected in the injection port of an Agilent 7890 gas chromatograph (GC) equipped with flame-ionization detection (FID) (Agilent, Las Rozas de Madrid, Spain). The GC conditions were as follows: 200 and 300 °C were the injector and detector temperatures, respectively; Agilent HP5 column (30 m, 0.32 mm diameter, 0.25 µm) with a 15 mL/min constant flow of He and 5:1 split; oven at 40 °C for 3 min, 10 °C/min to 80 °C, 40 °C/min to 200 °C and 4 isothermal min at 200 °C. The response of the GC was previously calibrated for each VOC by injecting known amounts. Vials without adsorbent were also included as controls. The amount of VOC retained by the adsorbents was indirectly calculated through the difference between the concentrations of VOCs in samples with adsorbents and controls. The results were calculated following Equation (1) and expressed as partition coefficient (*K*) values:(1)$${\left[VOC\right]}_{adsorbent}\;\left(\frac{g}{g}\right)=\frac{\left[\left({\left[VOC\right]}_{headspace}\;{\left(\;\frac{g}{\mathrm{mL}}\;\right)}^{control}-{\left[VOC\right]}_{headspace}\;{\left(\;\frac{g}{\mathrm{mL}}\;\right)}^{sample}\right)\cdot 22\;\mathrm{mL}\right]}{50\cdot 10^{-3}\;g};\;K=\frac{{\left[VOC\right]}_{adsorbent}\;\left(\frac{g}{g}\right)}{{\left[VOC\right]}_{headspace}\;\left(\;\frac{g}{\mathrm{mL}}\;\right)}$$

#### 2.2.2. Analysis of Retention Capacity of Adsorbents

The three substances with best performances as VOC adsorbents were selected and tested via their exposure to 100 µg of a single VOC in vapor state. The retention of VOCs by increasing amounts of pure adsorbents was analyzed (approximately from 30–50 mg). The sorption was analyzed by GC, following the same procedure described in Section 2.2.1.

### 2.3. Assessment of the Oxygen Scavenging Activity of the TM Product

The oxygen scavenging activity of these two types of films, a 50 μm thick LDPE-based monolayer containing the TM oxygen scavenger at a 10%wt. concentration and a 30 μm 3-layer (10/10/10) LDPE film with the center layer containing 10%wt. scavenger were analyzed in order to confirm their efficiency. The oxygen concentration in the headspace was measured over time by a non-invasive method based on phosphorescence quenching of a sensor dye. An Oxy-4 multichannel oxygen meter equipped with 4 polymer optical fibers and the Measurement Studio 2 software (PreSens, Regensburg, Germany) were used. In 250 mL wide-mouth glass bottles with glass stoppers, calibrated oxygen sensor spots PSt3 (Presens) were adhered on the inner wall of the bottle, and a film sample (1.2 g and 3.6 g of monolayer and 3-layer films, respectively) was introduced in the bottle and was immediately closed with vacuum paste. Adapters for round containers were set on the outside of the bottles fixing the optical fibers for sequential readings.

### 2.4. Analysis of the Capacity of Retention of VOCs Released by TM-Containing Films by Selected Adsorbents

The principal released VOCs by these films were identified and their retention capacity by the adsorbents selected in Section 2.2 was tested alone and in combination in order to study their synergistic/antagonistic effect. Adsorbents were exposed in a closed container to the VOCs delivered to the headspace by the oxygen scavenging 3-layer 10/10/10 µm film and their adsorption capacities were tested. The 3-layer film was selected because the potential application on food packaging would be more similar to this system because the oxygen scavenger components should not be in direct contact with the food. The amount of film was ca. 250 mg, and approx. 10 mg of every adsorbent was used on every test.

In a 22 mL vial, weighted amounts of the specific adsorbent or adsorbents were introduced. Then, pieces of the active film were weighed and included in the vial, which was hermetically sealed with an aluminum cap and rubber/PTFE septum. The samples were allowed to equilibrate during 72 h. Then, a carboxen/polydimethylsiloxane CAR/PDMS Solid Phase Microextraction (SPME) fiber was introduced in the vial and allowed to adsorb for 20 min. The SPME fiber was introduced in the injector of a gas chromatograph for the desorption of all VOCs. The GC was equipped with a mass detector and the compounds were identified using the NIST (National Institute of Standards and Technology) library. Samples without adsorbent were also included as controls.

The GC conditions were as follows: 250 °C the injector temperature; Agilent HP-5MS (5% phenyl methylpolysiloxane) column (30 m, 0.25 mm diameter, 0.25 µm) with a 0.92 mL/min constant flow of He and 5:1 split; oven at 40 °C for 4 min, 10 °C/min to 220 °C and 10 isothermal min at 220 °C. The conditions of the MS detector were: 1000 gain factor, voltage of 1692 V, frequency at 3.1 scans/s, step size at 0.1 m/z, MS source at 230 °C and MS quad at 150 °C. For each VOC found, a characteristic ion of the substance’s MS spectra was selected, and the number of those characteristic ions was monitored to determine the adsorbent efficiency. No calibration was carried out. The percentage of VOCs retained by the adsorbent was indirectly calculated by the difference between the samples with adsorbents and the controls.

### 2.5. Development of Films including Adsorbents

LDPE formulation compounded with 5 wt.% of Sylysia and 5 wt.% of TegoSorb was prepared by melt extrusion processing using a twin-screw extruder Coperion ZSK 26 Mc (Coperion, Stuttgart, Germany). LDPE pellets and Sylysia were fed through the main hopper of the extruder while Tegosorb was introduced inside the extruder through a side feeder. The temperature profile along the 10-barrel zones from hopper (zone 1) to die (zone 10) was 185–195–200–205–200–200–200–205–205–205 °C, the applied screw rotation speed was set at 550 rpm, and the throughput was 12 kg/h. Then, the compounded sample in pellet form was transformed into a film of around 35 µm using a single-screw extruder Brabender Stand-alone KE 30/32 (Brabender^®^ GmbH & Co. KG, Duisburg, Germany) equipped with an extrusion roller calender line. The temperature profile along the 6-barrel zones from hopper (zone 1) to die (zone 6) was 185–185–190–190–200–200 °C, screw speed 50 rpm and a roll speed of 14 m/min.

Their activities to scavenge released VOCs by TM-including films were analyzed as follows. In a 22 mL vial, a weighted mass of TM-monolayer film (Section 2.3) was included in the vial and then the same weight (“one side”), twice the weight (“two sides”), and four times the weight (“double two sides”) of the VOC-scavenger film were included in vials and closed. After 5 days, the headspace of the vials was analyzed by GC-MS. The sample named as “one side” represented a potential 3-layer film containing one pristine PE layer, the TM layer and the VOC scavenging layer. The “two sides” sample represented a potential 3-layer film containing a VOC scavenging layer, the TM layer and VOC scavenging layer. The “Double two sides” sample represented a potential 3-layer film containing VOC scavenging layer, the TOR layer and the VOC scavenging layer but with double the content of Sylysia and Tegosorb adsorbents. The GC conditions of analysis were analogous to Section 2.3.

### 2.6. Development and Efficiency of Oxygen Scavenger Packaging Systems including TM-Active Agent and VOC Adsorbers

The 10-layer structure was developed in three different steps by employing a coextrusion line equipped with three single-screw extruders (Dr Collin E30P, 25 L/D, COLLIN Lab & Pilot Solutions GmbH, Maitenbeth, Germany). The equipment has a feed-block that can provide up to five layers (ABCBA), and different multilayer rearrangements (e.g., ABC, BCB, ACA), bilayer (AC, BC) or monolayer structures. The first step consisted of the preparation of a 5-layer LDPE/tie/EVOH/tie/LDPE coextruded structure with an overall thickness of 33–36 µm. The processing parameters to obtain the symmetrical five layer “ABCBA” structure are displayed in Table 1. The chill roll temperature and the line speed were set at 60 °C and 20 m/min, respectively.

Thereafter, a coextrusion lamination of a PP/mPE bilayer structure over the 5-layer system was carried out, leading to a final overall thickness of 53–60 µm after this stage. Thus, a 7-layer structure as described in Table 2 was obtained. Finally, the third stage consisted of the coextrusion lamination of three-layer structure LDPE-VOC/LDPE + 33% TOR/LDPE-VOC over the 7-layer structure obtained previously to obtain a final multilayer system of 10 layers. The processing parameters of this stage to obtain the three-layer structure “ACA” extrusion-laminated over the 7-layer substrate are described in Table 3. The chill roll temperature and the line speed were set at 90 °C and 22 m/min, respectively, and the final dimensions of the 10-layer (PE-VOC/PE-TM/PE-VOC/LDPE/tie/EVOH/tie/LDPE/mPE/PP) structure were 300 mm width and 85 µm thickness, approximately. A similar structure was prepared by extruding LDPE without adsorbents in A to prepare a control with TOR but without VOC adsorbents (PE/PE-TM/PE/LDPE/tie/EVOH/tie/LDPE/mPE/PP).

The final thickness distribution of the 10-layer structure was evaluated by optical microscopy in transmittance mode using a Leica DM/LM optical microscope.

An amount of 1 g of the 10-layer film was hermetically closed in a 22 mL vial with air. After 5 days, an analysis of the vial headspace was conducted to compare the amount of volatile compounds released from films. A similar set-up was prepared with the control 10-layer film. The percentage of VOCs retained by adsorbents was evaluated.

### 2.7. Statistical Analysis

Analysis of variance (ANOVA) and Fisher’s multiple range test of the DSC and mechanical parameters, permeability, and overall migration data were performed using Statgraphics Plus 5.1. The number of replicates of the ANOVA analysis was stated in the experimental procedure for each test. Statistical comparisons were made from a randomized experimental design with a confidence level of 95%. Results were reported as the mean and standard deviation. A *p*-value less than 0.05 indicated that the mean values were significantly different between the samples.

## 3. Results and Discussion

### 3.1. Adsorption Capacity of VOC Adsorbents

As mentioned in the introduction, unsaturated polyolefins are excellent oxygen scavenger materials but, during the oxygen reaction process, chain scission occurs releasing low-molecular-weight oxidized organic compounds including organic acids, aldehydes and ketones. This is what occurs when TOR containing films are exposed to atmospheres containing oxygen. Thus, in order to reduce the release of VOCs into package headspace, several adsorbent materials were exposed to individual volatile compounds to check their adsorption capacity. The selected volatiles were mainly short chain organic acids and aldehydes. Also, acetone, amyl formate and cyclooctane were selected as representative compounds of the ketone, esters and hydrocarbon families.

Figure 1 shows the partition coefficients between adsorbents and headspace for all tested volatile organic compounds. Silica is a solid form of silicon dioxide and is specially synthesized to be highly porous. Although SiO_2_ is a polar compound, the surfaces of functionalized silica are usually covered with a layer of hydroxyl groups which can be largely substituted through treatment with suitable reagents to reduce polarity [14]. Adsorption results of Sylysia SY350 (Syl) in this first adsorption analysis evidenced this hydrophilic silica as a potential adsorbent to scavenge VOCs from TOR byproducts. This silica presented *K* values above 1000 for carboxylic acids, aldehydes, ketones and esters, although *K* values were lower for cyclooctane due to the increased apolar character of this alkane. On the contrary, the hydrophobic silica Dumacil 100 FG (Duma) did not present relevant adsorption values for any of the tested organic substances. Previous works have already shown the effects of physical structure and chemical modification of adsorbents on the adsorption performance [15,16].

Tego active agent, a product from Evonik, is a masterbatch of LDPE-containing zinc ricinoleate, the zinc salt of the major fatty acid found in castor oil often used as an odor-adsorbing agent [17,18]. The most relevant *K* values presented by this product were towards organic acids, but relevantly towards more apolar compounds and those with higher molecular weights, such as cyclooctane.

Cyclodextrins (CDs) are cyclic oligosaccharides which have been widely used as scavengers of organic compounds of low polarity. They present a truncated cone shape with a polar outer surface and an apolar inner surface [19,20]. Among the diverse CDs, βCD has been reported to complex several types of organic compounds. This CD has also been functionalized to improve their compatibility with polymers or water solubility. This is the case of hydroxypropyl-βCD (HPβCD), obtained through the functionalization of hydroxyl by hydroxypropyl groups, and commonly used for the complexation of poorly soluble drugs [21,22]. In general, both CDs presented acceptable *K* values, with the retention capacity of both tested CDs being very similar.

Ultiloc 5003 (Ultiloc) is a novel copolymer of vinyl alcohol and vinyl amine that was selected as a potential aldehyde scavenger since aldehydes are known to react with primary amines to produce Schiff bases [23,24], and these aldehydes, especially C5 to C8, are known to promote rancid off-flavor which can affect sensory properties in food packaging applications. Nevertheless, a low adsorption capacity was observed for polyvinyl alcohol/polyvinyl amine Ultiloc.

Finally, Tenax, a modified polyphenylene oxide and powerful adsorbent commonly used for the trapping of VOCs in water samples, or as food simulant of solid foods in migration tests, did not present the expected results for most VOCs, although its adsorption capacity towards molecules with higher molecular size, such as hexanoic acid, amylformate and cyclooctane, was quite high.

From this first analysis, Syl, Tego and βCD were the adsorbents with better performances, and particularly, Syl was the material that presented the greatest adsorption capacity on most VOCs; Tego seemed to provide good adsorption properties to organic acids and alkanes but showed a lower capacity with aldehydes, and βCD presented an average adsorption capacity to all compounds and their incorporation in polymers can be performed easily through extrusion rather than HPβCD. Tenax (60–80 mesh), due to its high cost, low availability and large particle size, was discontinued for further tests.

The effect of increasing the amount of adsorbents with the best performances on VOC retention was also analyzed to assess whether the retention capacity measured is a consequence of a partition or is an exhaustion of the adsorption capacity. Figure 2 shows two examples of the results obtained for the capacity of increasing amounts of Syl, Tego and βCD in retaining acetic acid, and valeraldehyde. As can be seen, *K* values for the tested VOCs and the three adsorbents did not present any trend with respect to the amount of the solid sample included in the analyses. Although some variations were observed, in general, it can be concluded that partition equilibria of the VOCs between the adsorbent and the headspace were achieved. These results also confirmed that Syl retained relevant amounts of most polar VOCs thanks to its hydrophilic character but failed to retain alkanes and alkenes. Tego was good for acids but performed greatly with hydrocarbons. βCD always presented intermediate *K* values, being, therefore, discarded for the following trials.

### 3.2. Oxygen Scavenging of TOR-Containing Films

Two LDPE films containing oxygen scavenger TM were successfully developed through melting extrusion: a 50 μm thick LDPE based monolayer containing TM at 10%wt. and a 30 μm 3-layer (10/10/10) LDPE film with the center layer containing 10%wt. scavenger. Both films were supplied by Evonik in aluminum/PE pouches under vacuum.

First, the oxygen scavenging activities of these active films were analyzed using Oxi-4 Presens equipment. Oxygen scavenging activities were periodically measured for 5 days and their resulting kinetics are graphed in Figure 3.

As Figure 3a shows, both oxygen scavenger films presented a similar kinetic profile and final oxygen scavenging activities. At first, the monolayer showed slightly faster scavenging activity probably due to the easier accessibility of TM scavenger to oxygen, but, in general, oxygen scavenging activities of both films were not relevantly different, probably due to the highly oxygen-permeable characteristic of LDPE. The final data of the oxygen scavenging capacities were 351 ± 9 cm^3^ and 349 ± 17 cm^3^ per gram of active agent incorporated into the film for the monolayer and the 3-layer, respectively, values obtained after 5 days of exposition to air. Assuming the 3-layer 10/10*/10 µm structure with the scavenger at 10%wt. load, the scavenging capacity will be 316 ± 8 cm^3^/m^2^. These values are excellent when compared to other potential scavengers: polybutadiene with cobalt salt scavenge of 15 mg/100 mg of polymer (0.2 mL/g) [12,25], iron-based sachets, ca. 45 mL/g, 2.5 mL/g of an iron–kaolinite composite in LDPE [26], or polymer composites including TiO_2_ cs. 30 mL/g [27].

Figure 3b showed that the scavenging activity of the 3-layer active film was slightly slower and lower at lower temperature (7 °C), and higher deviation values between samples were found. The increase in oxygen scavenging activity with the temperature has also been observed when using antioxidants as oxygen scavenger, such as gallic acid and tea polyphenols, because their activities increased at higher temperatures [28,29].

### 3.3. VOC Retention of Selected Adsorbents in Real Condition Testing

The use of polyalkenes for oxygen scavenging activity results in the generation of oxidation products from allylic carbon hydrogen bonds oxidative degradation that can turn out a problem due to their pungent odor of these byproducts, referred in this work as Volatile Organic Compounds (VOCs) [9,30]. Therefore, the capacity of adsorbents selected in Section 3.1 to retain VOCs released by the active 3-layer films containing TM was analyzed in order to diminish this secondary effect.

First, the main volatile compounds released from TM were identified by gas chromatography mass spectrometry using the NIST library. The corresponding compounds were quantitatively measured attending to their most relevant ions. Table 4 presents the identified compounds, their retention time, and the ion mass utilized for quantification.

They all were quantified and their adsorptions in both pure and mixed adsorbents were analyzed after 72 h of exposure. Due to their amount and their sensory impact, carboxylic acids and esters of formic acid are of paramount importance. Gas chromatograms of samples with Sylysia, Tegosorb adsorbents and their mixture are compared to the chromatogram of the control sample without adsorbent substances in Figure 4.

The adsorbent retention percentages of most important VOCs released by the 3-layer film are presented in Figure 5. In general, most identified VOCs were relevantly retained by the adsorbents (Figure 5). VOC adsorption depends on various factors, such as VOC type and concentration, and the interactions between the adsorbent substance and VOCs that are mainly governed by their corresponding polarities [31,32].

As can be seen in Figure 5, the adsorbents, individually or mixed, retained a large part of the carboxylic acids released by the scavenger. Sylysia retained above 80% of them with the exception of formic acid. Tegosorb performed above 80% for these compounds and above 95% for formic acid. Vials with both adsorbents performed in general better than individually. Also, the scavengers performed well with esters of formic acid, especially Sylysia or the combination of both adsorbents.

Several alkanes from C5 to C9 were identified in low amounts and were scarcely retained by adsorbents either individually or mixed, with the exception of cyclohexane. Three alcohols were present in the vial headspace at low concentrations, according to the small peaks observed in the chromatograms. The alcohol retention by Tegosorb was very limited. On the contrary, Sylysia or the mixture retained ca. 60% of butanol, and ca. 80% for C5-alcohols. Other works have also demonstrated various silica as efficient adsorbents for various VOCs’ removal thanks to their uniform and open-pore structures [33,34]. Similar comments are appropriate for retention capacity of adsorbents on aldehydes and ketones Their retention, especially in vials containing Sylysia was found to be between 60 and 80% (Figure 5). Considering the large retention observed for these compounds when tested as single contaminants (Figure 1), the preferential retention of other compounds, present in large quantities (acids and esters) could have reduced the scavenging of aldehydes and alcohols.

Some studies have evidenced carbon sorbents quantitively trapped in a wide range of VOCs from C3 to C12 whereas mesoporous silica trapped considerably larger molecules from C8 to C12 with the potential to go beyond C12 [32]. This was confirmed by the great retention of Syl and their mixtures on VOCs with high molecular weight (Figure 5).

After these results, a mixture of Syl and Tego at equal concentrations was selected for incorporation in a polyethylene film and confirmed their capacity to retain VOCs when included in a polymer matrix.

### 3.4. VOC Retention of Developed TM Adsorbents Prototypes

Figure 6 shows the results obtained comparing the content of 15 selected compounds in control samples (only the TM film) with the other samples including films containing the adsorbents at a 5%wt. concentration. The sample named “One side” included the same weight of film than the oxygen scavenger monolayer to be simulated as a trilayer in which one of the external side layers contains the VOC adsorbents. The “Two sides” sample included twice the weight of the oxygen scavenger film to simulate as a trilayer in which the oxygen adsorber layer is sandwiched between two layers of VOC scavenger LDPE. Finally, “Double two sides” included double quantity of VOC adsorbents film to simulate a trilayer like the “Two sides” but with twice the concentration of Sylysia and Tegosorb. As can be seen, the VOC scavenging film systems retained most of the released compounds, achieving percentages of VOC retained values between 85 and 100%. As was expected, the designed system improved the retention as the amount of adsorbents increased, although no significant differences were observed. The “Two sides” samples presented slightly higher VOC retention than “One side” samples. Therefore, the design of a system including a trilayer of LDPE containing 10%wt. TM in the central layer and two identical external layers containing 5%wt. of Sylysia and Tegosorb appeared to be sufficient to reduce the risk of sensorial damage on packaged products caused by the release of low-molecular-weight compounds produce by oxidative reactions of TOR.

### 3.5. VOC Retention of Developed Multilayer Active Systems

As described in Section 3.3, it was evidenced that the best performance of VOC retention occurred at both sides, and therefore, a multilayer system was designed in which the layer containing TM was sandwiched between two layers of PE containing the adsorbent. The multilayer system PP/tie/EVOH/tie/PP/PE-VOC/PE-TM/PE-VOC was successfully developed. PE-VOC corresponded to PE layer including the adsorbents at 5%wt. An EVOH layer was included in order to protect the PE-TM layer from outer oxygen. Other oxygen scavenger developments were also based on the design of multilayer systems including one or more layers which are permeable to oxygen between the food product and the oxygen scavenger layer and an oxygen barrier layer towards the outside of the package [10]. The final 10-layer films were evaluated by optical microscopy in transmittance mode, to determine the thickness of each layer, with an accuracy of 3 µm. Figure 7 shows an image of the film with the thickness of the different layers, the right layers correspond to the PE-VOC/PE-TM/PE-VOC.

As can be seen in Figure 8, most of the volatile compounds were reduced by the inclusion of the VOC scavengers, especially alkanes, acids, ketones and esters. Alkanes, as cyclohexane and trimethylpentane, and ketones, such as 3-hexanone and 2-hexanone, displayed retention values higher than 95%. Organic acids, such as formic, acetic and propanoic acids, evinced intermediate retention values between 50–60% with respect to the control released amount. On the contrary, alcohols, such as butanol and pentanol, and hexanal were apparently released with the same intensity from the film with VOC scavengers. Nevertheless, these compounds were present as a trace, and were difficult to identify, and therefore, scarcely relevant as compounds with potential sensory effects on packaged products.

## 4. Conclusions

This work has presented the scientific and technological steps for the development of an active packaging with oxygen scavenger capacity. This study has included the study of the efficiency of the TOR oxygen scavenger (TM) films, the formation of volatile organic compounds (VOCs) generated by the TM active compound in the presence of oxygen, the search for VOC adsorbers, and the development of a multilayer film that included both the oxygen scavenger and VOCs adsorbents. First, an optimal product including TOR was prepared as a masterbatch blended with LDPE (among other polyolefins), that was further diluted to a preferred used concentration of ca. 10%wt. Subsequently, the scavenging material was incorporated in a multilayer system that included a high barrier layer to avoid the reaction with the oxygen from the external atmosphere. Also, a functional layer was required to impede direct contact of the scavenger material with the food product.

Among the advantages of the TOR masterbatch, their extraordinarily high scavenging capacity above 350 mL of oxygen per gram of material, their fast kinetics (80% of its capacity in 5 days), and the independency of this activity with humidity were highlighted. Sylysia, a hydrophilic silica, and Tegosorb 88, a product based on zinc ricinoleate, have demonstrated a good retention of VOCs. TOR product presented efficient oxygen scavenger values at low and room temperatures. Several VOCs were generated as byproducts of the oxodegradation of this oxygen scavenger in the presence of oxygen. The incorporation of layers including absorbent substances have demonstrated the efficiency on VOC retention. The design and the incorporation of this layer was successfully carried out in a multilayer system that can be shaped into sheets, containers or other packaging materials through thermal forming technologies, such as melting extrusion or compression/blow molding. The main advantage of this kind of oxygen scavenging system lies with the fact that it can be used to package dry food products sensitive to oxidation, from nuts to fish rich in unsaturated fatty acids such as salmon.

## Figures and Tables

**Figure 1 polymers-15-03899-f001:**
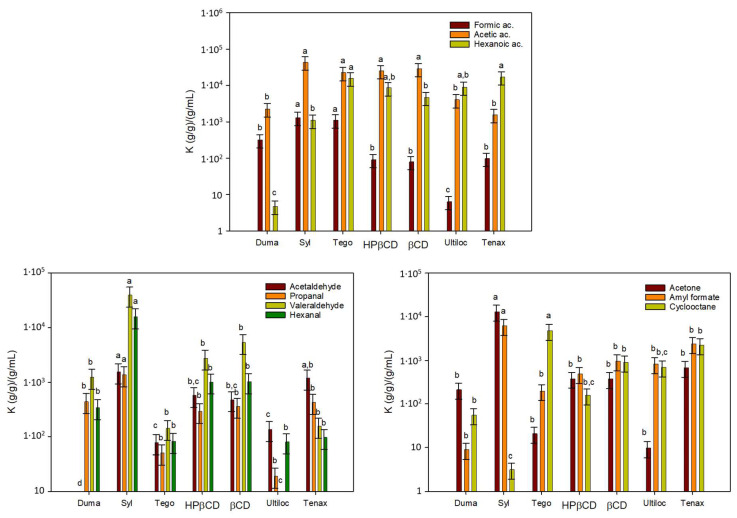
Partition coefficients (*K*) for the selected volatile organic compounds between the tested adsorbents named in the *x*-axis and the headspace. “a, b, c and d” indicate significant differences among the *K* values of each organic compound between adsorbents.

**Figure 2 polymers-15-03899-f002:**
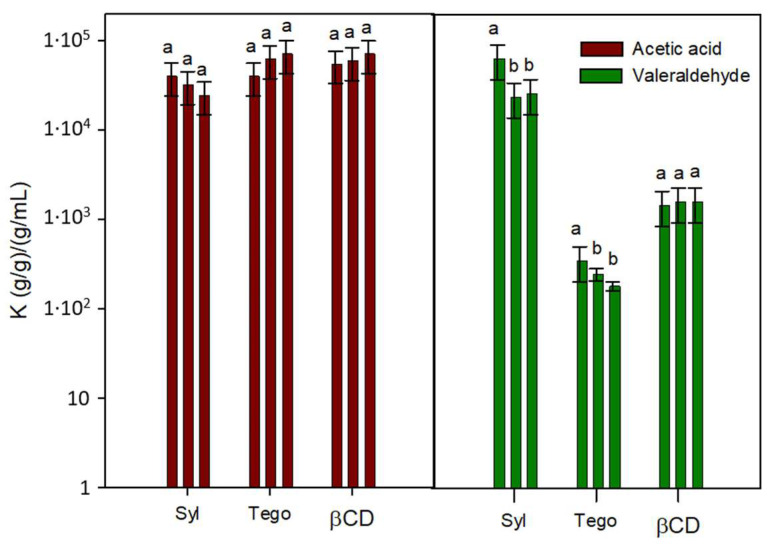
Partition coefficients (*K*) for acetic acid and valeraldehyde between Syl, Tego and βCD adsorbents and the headspace (the amount of adsorbent increases from left to right for each adsorbent and VOC). “a and b” indicate significant differences among the *K* values of each organic compound between adsorbents.

**Figure 3 polymers-15-03899-f003:**
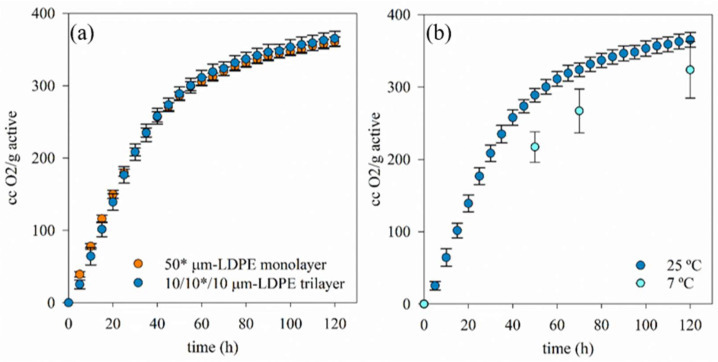
Kinetic oxygen scavenging activities of (**a**) both mono (50 µm) and trilayer (10/10*/10 µm) active films at room temperature; (**b**) and trilayer films at room temperature and 7 °C. Note: (*) indicates the layer containing TM.

**Figure 4 polymers-15-03899-f004:**
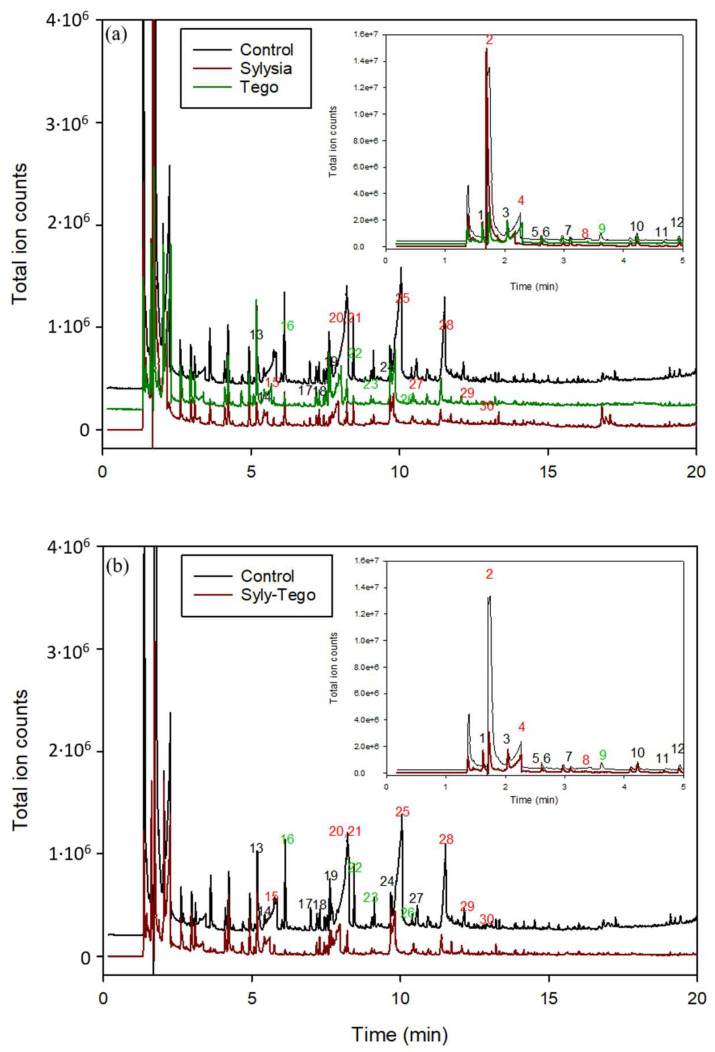
Representative GC-MS chromatograms from headspace of vial containing (**a**) black line, the active trilayer film (control); red line, the vial containing the active film and Sylysia; and green line, the vial containing the active film and Tegosorb (Tego); (**b**) black line, the active 3-layer film (control); red line, the vial containing the active film and mixture of Sylysia and Tegosorb. Numbers in figures indicate the identified compound according to Table 4.

**Figure 5 polymers-15-03899-f005:**
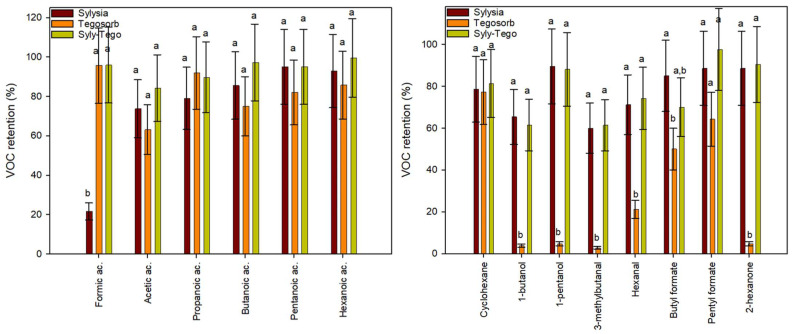
Kinetic retention of VOCs by adsorbents Syl, Tego, and their mixture after 72 h in air at room temperature of exposure to 3-layer films of organic acids (**left**); other VOCs (**right**). “a and b” indicate significant differences among VOC retention values of each organic compound between adsorbents.

**Figure 6 polymers-15-03899-f006:**
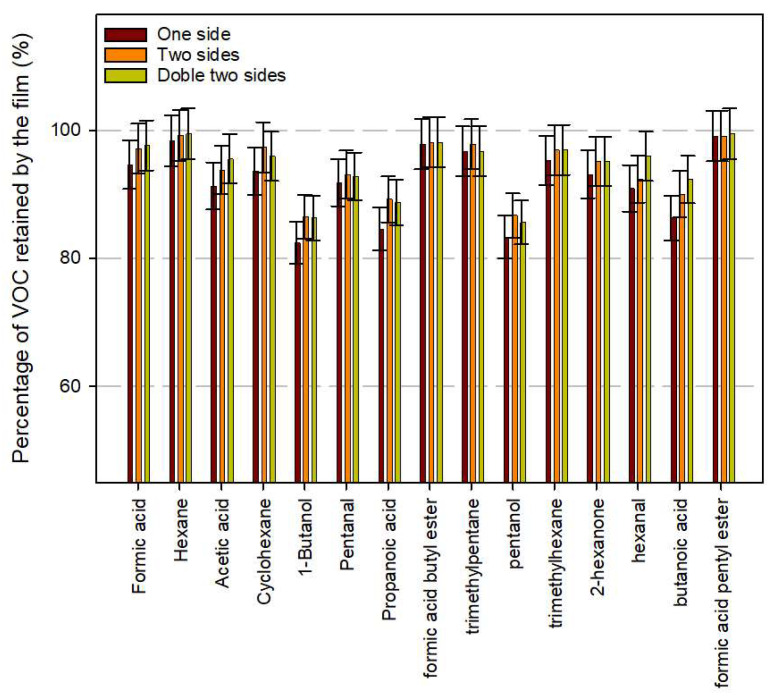
Retention of some volatile compounds released from TM-containing films containing VOC scavengers.

**Figure 7 polymers-15-03899-f007:**
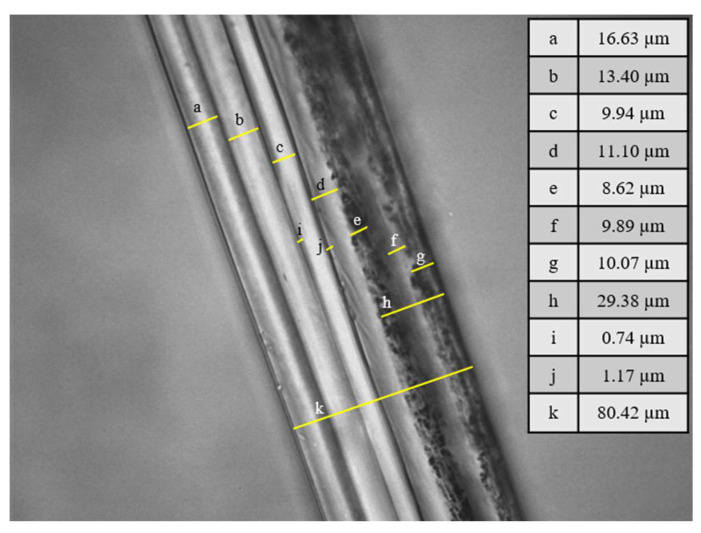
Thicknesses of the diverse layers of the multilayer structure developed. Left, PP layer; right, the PE-VOC layer.

**Figure 8 polymers-15-03899-f008:**
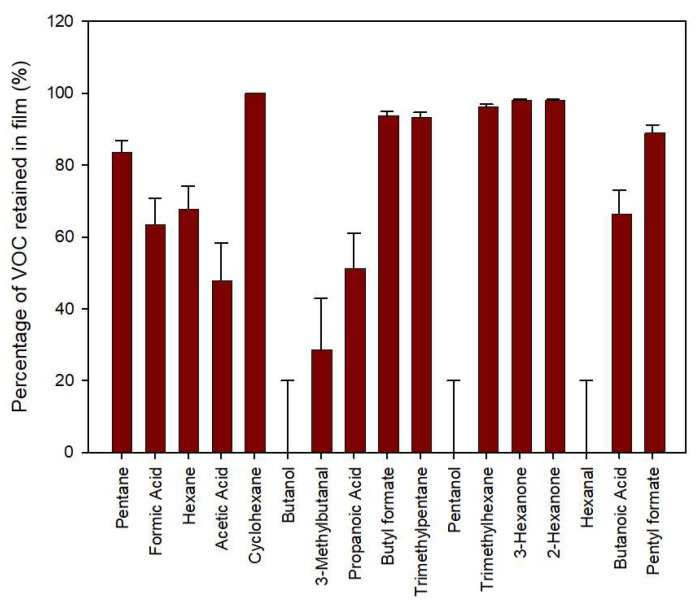
Percentage of diverse VOCs retained by active film with oxygen scavenger and VOC scavenger (PP/tie/EVOH/tie/PP/PE-VOC/PE-TM/PE-VOC) related to the concentrations measured in vials with film containing only oxygen scavenger (PP/tie/EVOH/tie/PP/PE/PE-TM/PE) after 5 days of storage at room temperature.

**Table 1 polymers-15-03899-t001:** Parameters and scheme of the first step of the multilayer development: a LDPE/tie/EVOH/tie/LDPE film structure.

	Screw Speed (rpm)	T1 ^1^ (°C)	T2 ^1^ (°C)	T3 ^1^ (°C)	T4 ^1^ (°C)	T5 ^1^ (°C)	COEX (°C)	DIE (°C)
Extruder A	120	165	180	180	180	180	210	200
Extruder B	25	165	175	175	180	185		
Extruder C	55	205	220	220	220	220		
Scheme	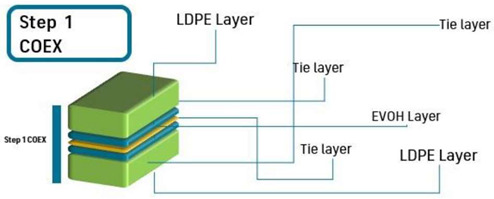							

^1^ T1 to T5 are the five heating zones of the extruder screw.

**Table 2 polymers-15-03899-t002:** Parameters and scheme of the second step of the multilayer development: PP/mPE coextrusion lamination onto LDPE/tie/EVOH/tie/LDPE.

	Screw Speed (rpm)	T1 ^1^ (°C)	T2 ^1^ (°C)	T3 ^1^ (°C)	T4 ^1^ (°C)	T5 ^1^ (°C)	COEX (°C)	DIE (°C)
Extruder A							210	200
Extruder B	5	190	215	225	240	245		
Extruder C	80	210	240	240	240	240		
Scheme	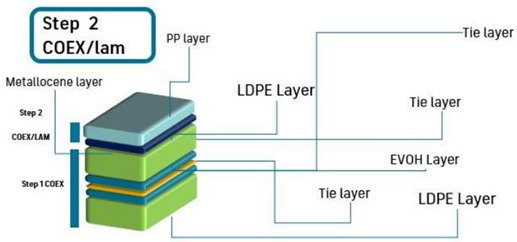							

^1^ T1 to T5 are the five heating zones of the extruder screw.

**Table 3 polymers-15-03899-t003:** Parameters and scheme of the third step of the multilayer development: PE-VOC/PE-TM/PE-VOC coextrusion lamination onto LDPE/tie/EVOH/tie/LDPE/mPE/PP.

	Screw Speed (rpm)	T1 ^1^ (°C)	T2 ^1^ (°C)	T3 ^1^ (°C)	T4 ^1^ (°C)	T5 ^1^ (°C)	COEX (°C)	DIE (°C)
Extruder A	120	165	190	190	195	200	205	205
Extruder B	-	-	-	-	-	-		
Extruder C	100	190	190	190	195	200		
Scheme	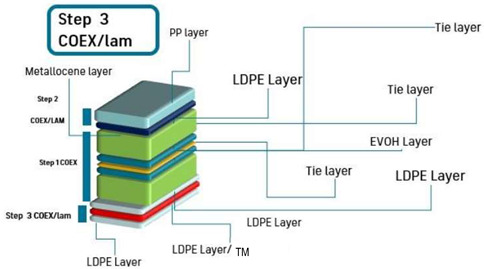							

^1^ T1 to T5 are the five heating zones of the extruder screw.

**Table 4 polymers-15-03899-t004:** Compounds tentatively identified by GC-MS using NIST library, their retention time, and the ion mass (uma) utilized for quantification.

Peak	Retention Time (min)	Volatile Compound	Ion Mass	Peak	Retention Time (min)	Volatile Compound	Ion Mass
1	1.63	pentane	43	17	6.98	2-pentenol	68
2	1.73	formic acid	46	18	7.63	2-heptanone	43
3	2.05	hexane	57.1	19	7.69	cyclohexanone	55
4	2.24	acetic acid	60	20	8.1	pentanoic acid	60
5	2.65	cyclohexane	84.1	21	8.23	cyclooctane	56
6	2.69	butanol	56	22	8.44	hexyl formate	56.1
7	3.11	3-methylbutanal	57	23	9.13	cyclohexyl formate	67
8	3.4	propanoic acid	74	24	9.73	2-octanone	43
9	3.62	Butyl formate	56.1	25	9.96	hexanoic acid	60
10	4.12	trimethylpentane	71.1	26	10.41	heptyl formate	70.1
11	4.72	pentanol	55	27	10.53	cyclopentyl carboxilic acid	73
12	4.94	trimethylhexane	57.1	28	11.48	heptanoic acid	60
13	5.1	3-hexanone	57	29	12.16	cyclohexane carboxilic acid	55.1
	5.19	2-hexanone	43	30	12.92	octanoic acid	60
14	5.44	hexanal	56	31	16.18	tetradecanal	82.1
15	5.75	butanoic acid	60	32	16.93	1-phenyl 1-hexanone	105
16	6.13	Pentyl formate	70.1				

## Data Availability

Data will be available upon request.

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
