# Peer review of "Designing an Oxygen Scavenger Multilayer System Including Volatile Organic Compound (VOC) Adsorbents for Potential Use in Food Packaging"

_polymers, 2023, doi:10.3390/polym15193899_

Round 1

Reviewer 1 Report

The study is interesting and is expected to be a good addition to the available literature. However, some minor comments are below.

1- Figures need to improved with proper labelling especially figure 6 is not very clear.

2- Introduction needs to be improved to clearly give the background knowledge.

3- conclusion needs to include the possible applications of the prepared films.

4- The language and grammar of overall manuscript needs to be improved.

 The language and grammar of overall manuscript needs to be improved.

Author Response

Comments and Suggestions for Authors

The study is interesting and is expected to be a good addition to the available literature. However, some minor comments are below.

  • Figures need to improve with proper labelling especially figure 6 is not very clear.

A: Figures have been revised and improved, and, following the recommendation of reviewer 2, statistical analysis was included in some figures.

2- Introduction needs to be improve to clearly give the background knowledge.

A: The authors agree with the reviewer. The introduction has been revised and improved. Grammatical tenses have been corrected, extra information has been included and some of the text was moved to conclusions.

3- conclusion needs to include the possible applications of the prepared films.

A: Conclusions was revised.

4- The language and grammar of overall manuscript needs to be improved.

A: Manuscript was revised.

Comments on the Quality of English Language

 The language and grammar of overall manuscript needs to be improved

A:The manuscript was revised and minor changes were included.

Reviewer 2 Report

The submitted manuscript entitled „Designing a oxygen scavenger multiplayer system including volatile organic compounds (VOC)-adsorbents for potential use in food packaging“ is devoted to the study of oxygen scavengers incorporated in multiplayer polymer systems.  It is focused on the analysis of different adsorbents for volatile organic compounds.

I have some minor comments that are attached below. However, the most important issue is related to Supplementary Material, which is discussed in the text, but it was not included in the file. Due to this reason, I was not able assess the results part concerning VOC retention in real condition testing (Chapter 3.3). These files should be added to the manuscript so that it can proceed further.

Comments:

·         I am not sure why some materials are not included in 2.1. Materials section (e.g. PP, EVOH).

·         I miss the results (significant differences) of statistical analysis in the tables and graphs.

·         Right parenthesis should be added in Figure 4 title (at the end).

·         Labels of graph axes should be unified (either small or capital letters…)

·         “doble” on line 474 should be rewritten to “double”

·         Figure 6 – description of individual layers (thickness) should be in bigger letters, as well as the scale of the image is not readable.

·         line 510 – word “more” is mentioned twice inappropriately.

·         Line 529 – a gap should be added between by AND products.

I do not find any serious issues in English Language style, instead of minor comments included in an appropriate section.

Author Response

Comments and Suggestions for Authors

The submitted manuscript entitled „Designing a oxygen scavenger multiplayer system including volatile organic compounds (VOC)-adsorbents for potential use in food packaging“ is devoted to the study of oxygen scavengers incorporated in multiplayer polymer systems.  It is focused on the analysis of different adsorbents for volatile organic compounds.

I have some minor comments that are attached below. However, the most important issue is related to Supplementary Material, which is discussed in the text, but it was not included in the file. Due to this reason, I was not able assess the results part concerning VOC retention in real condition testing (Chapter 3.3). These files should be added to the manuscript so that it can proceed further.

 A: Authors agree with reviewer and Sections 2.4. and 3.3. were completely revised and improved. A Table and a Figure from Supplementary Material was incorporated in the manuscript. Thus, the revised manuscript do not contain Supplementary file.

Comments:

  • I am not sure why some materials are not included in 2.1. Materials section (e.g. PP, EVOH).

A: This information was included in this revised manuscript.

  • I miss the results (significant differences) of statistical analysis in the tables and graphs.

A: Statistical analysis were included.

  • Right parenthesis should be added in Figure 4 title (at the end).

A: Added (new Figure 5), thank you.

  • Labels of graph axes should be unified (either small or capital letters…)

A: Revised.

  • “doble” on line 474 should be rewritten to “double”

A: Corrected

  • Figure 6 – description of individual layers (thickness) should be in bigger letters, as well as the scale of the image is not readable.

A: Figure 6 (Figure 7 in the revised manuscript) was improved.

  • line 510 – word “more” is mentioned twice inappropriately.

A: Corrected

  • Line 529 – a gap should be added between by AND products.

A: Corrected

Comments on the Quality of English Language

I do not find any serious issues in English Language style, instead of minor comments included in an appropriate section.

Reviewer 3 Report

The reviewed manuscript explores the results of research efforts aimed at developing a multilayer oxygen scavenging system using trans-polyactenamer rubber (TOR) for active food packaging applications. The authors conducted a comprehensive study to evaluate the adsorption capabilities of various materials, including zeolites, silicas, cyclodextrins and polymers, with the aim of identifying those with optimal properties for the retention of representative organic compounds. They also accurately identified the primary volatile organic compounds (VOCs) released from TOR-containing films and assessed their retention by both pure adsorbents and polyethylene containing selected compounds. The concentrations of these VOCs showed significant reductions, ranging from 10 to 100-fold, depending on the specific VOC.

The manuscript is well organised, clear and of considerable relevance to the field. It maintains scientific rigour and the experimental design appears to be well suited to the study.

The reference list contains 35 publications, 27 of which were published within the last five years. Visual aids such as graphs, images, diagrams and figures included in the document are easy to understand and do not pose challenges for interpretation.

The conclusions drawn in the manuscript are consistent with the evidence and arguments presented.

In my opinion, the manuscript is suitable for acceptance in its present form.

Author Response

Comments and Suggestions for Authors

The reviewed manuscript explores the results of research efforts aimed at developing a multilayer oxygen scavenging system using trans-polyactenamer rubber (TOR) for active food packaging applications. The authors conducted a comprehensive study to evaluate the adsorption capabilities of various materials, including zeolites, silicas, cyclodextrins and polymers, with the aim of identifying those with optimal properties for the retention of representative organic compounds. They also accurately identified the primary volatile organic compounds (VOCs) released from TOR-containing films and assessed their retention by both pure adsorbents and polyethylene containing selected compounds. The concentrations of these VOCs showed significant reductions, ranging from 10 to 100-fold, depending on the specific VOC.

The manuscript is well organised, clear and of considerable relevance to the field. It maintains scientific rigour and the experimental design appears to be well suited to the study.

The reference list contains 35 publications, 27 of which were published within the last five years. Visual aids such as graphs, images, diagrams and figures included in the document are easy to understand and do not pose challenges for interpretation.

The conclusions drawn in the manuscript are consistent with the evidence and arguments presented.

In my opinion, the manuscript is suitable for acceptance in its present form.

A: The authors are grateful for the reviewer's appreciation.

Round 2

Reviewer 2 Report

Authors have addressed the comments and the manuscript has been significantly improved, the missing files from Supplementary Material were included.

I have only minor comment regarding Figure 2. After this is appropriately edited, I recommend the submitted manuscript for publication in the Polymers Journal.

Figure 2 – a letter “c” is not included in the graph, thus it should not be in the figure legend.

Author Response

Reply is as a pdf file
